# Assessing Sub-Saharan Africa's readiness to address the impact of climate change and health: A scoping review

Aminata Kilungo[1]*, God'sgift Chukwuonye[2], Victor Okpanachi[1], Hussein Mohamed[3]

**1** Department of Community, Environment and Policy, University of Arizona, Tucson, Arizona, United States of America, **2** Department of Environmental Sciences, The University of Arizona, Tucson, Arizona, United States of America, **3** Department of Environmental and Occupational Health, Muhimbili University of Health and Allied Sciences, Dar es Salaam, Tanzania

* paminata@arizona.edu

## Abstract

Climate change severely threatens global public health, with sub-Saharan Africa (SSA) projected to experience profound impacts. This scoping review aimed to provide a comprehensive overview of current research on climate change and its health implications in SSA while identifying research gaps and outlining the necessary resources and policy interventions to strengthen public health resilience in the region. Literature was retrieved from four databases (PubMed, Scopus, Embase and Web of Science) using the keywords "climate change," "health," and "sub-Saharan Africa" and this study was conducted using the PRISMA framework. The inclusion criteria were peer-reviewed studies published in English between January 1, 2001, and August 1, 2024, that examined the effects of climate change in SSA, assessed its impacts on health outcomes,A total of 7851 journal articles were identified from the initial search, and after screening, 153 studies were included for review. The included studies were published between January 2001 and August 2024. Although extensive studies have been conducted on extreme heat (71 studies), drought (45 studies), extreme precipitation events (52 studies), and flooding (34 studies), important themes such as air quality (10 studies), chemical water quality (8 studies) and natural disasters (8 studies) have been understudied. Additionally, this scoping review revealed a geographical gap in climate change and health studies, as only 24 out of 53 countries in sub-Saharan Africa were represented. The key deficiencies identified include limited funding, technological constraints, inadequate climate policies, and a lack of community-focused adaptation plans. Moreover, this review highlights the urgent need for resilient healthcare systems capable of addressing climate-related health risks effectively. Addressing these gaps is essential for developing targeted strategies to mitigate climate change's health impacts and increase resilience in SSA communities. This review aims to inform policymakers, researchers, and stakeholders about critical areas requiring attention and investment by enhancing our understanding of

**Data availability statement:** All relevant data are within the paper and its Supporting Information files.

**Funding:** The author(s) received no specific funding for this work.

**Competing interests:** The authors have declared that no competing interests exist.

these challenges and gaps. Strengthening research capacities, fostering collaboration, and implementing evidence-based policies are imperative steps toward achieving sustainable health outcomes in the face of a changing climate in SSA.

## Introduction

Climate change poses a significant and escalating threat to public health globally, with particularly profound implications for vulnerable regions such as sub-Saharan Africa (SSA) and other low-income countries. Characterized by diverse climatic conditions and socioeconomic challenges, SSA faces heightened susceptibility to the multifaceted impacts of climate change [1–7]. Extreme heatwaves, shifting precipitation patterns, increased flooding frequency and intensity, and prolonged droughts directly influence health outcomes in the region [8–11]. These environmental shifts exacerbate existing health disparities and introduce new risks, impacting the well-being of more than one billion people [12] in SSA.

In SSA and other low-income countries, the impact of climate change will be more dire due to unaddressed structural and socioeconomic vulnerabilities, poor governance, and inadequate supportive policies to address climate change and health issues. These unaddressed issues have resulted in other challenges, including water insecurity, poor healthcare infrastructure, a lag in research and training needed to address climate issues, and others [13–15]. For instance, in places with a high burden of diarrhea disease due to a lack of Water, Sanitation, and Hygiene (WASH) services, climate change poses additional risks, as countries are now facing more diarrhea diseases and, in some areas, cholera outbreaks, most of which are attributed to climate change [2,16–20]. Health conditions that are expected to be exacerbated in addition to waterborne and water-related diseases include foodborne diseases, vectorborne diseases, mental health, high morbidity and mortality due to heat stroke, malnutrition, and many others associated with natural disasters. In these countries, we are already seeing overwhelmed healthcare systems that are not equipped to address these public health challenges [14,21,22].

Although some African countries have developed plans for adaptation and mitigation strategies as required by the UN [23], there is little evidence on whether and how these plans are implemented. In addition, limited research has been conducted in SSA to guide country-specific plans for adaptation and mitigation and other associated measures to address or reduce the impact of climate change on health [24,25]. Limited data is collected at the ground level to guide some of these plans and interventions. Most plans and research approaches are based on broader global perspectives [7,26–28]. Given that SSA has unique challenges, research to guide adaptation, mitigation, and resilience efforts must be unique and specific to the region, if not country-specific. Given these challenges, SSA needs to scale up research, education, climate change financing, and policy – all collectively, to move forward toward sustainable solutions for long-term climate change resilience. Most importantly, there is a need to set a road map for priorities to guide efforts, focusing on addressing challenges specific to the region.

To understand the research that has been conducted to guide some of these efforts, this scoping review, following the PRISMA framework, was conducted to examine current research on climate change and health impacts, policy, and to identify existing gaps in research and the need for policy-making to guide interventions to improve public health and climate change resilience in Africa. This scoping review aims to improve our understanding of where SSA is conducting research to generate specific data to address the complex challenges related to health outcomes due to climate change. The specific research questions guiding this review are as follows:

1. What are the geographic and thematic gaps in SSA's climate change and health research, and how do these gaps affect our understanding of the region's climate-induced health outcomes and vulnerabilities?

2. How do extreme weather events, such as heatwaves, droughts, and floods, interact with social determinants of health to influence health vulnerabilities and outcomes?

3. What practical solutions and community-based adaptation strategies can be developed and implemented to enhance resilience and mitigating health impacts to climate change

4. How are climate change and health studies funded?

## Method

A scoping review was conducted on PubMed, Embase, Web of Science and Scopus using the keywords "climate change," "health," and "sub-Saharan Africa" and imported to Covidence [29] to streamline and facilitate the review. The Covidence software follows the PRISMA approach for scoping review (Fig 1). A total of 7851 journal articles were identified using the keywords and inclusion criteria. No white papers, gray literature, review papers, or other non-peer reviewed sources were included. Only peer-reviewed primary research articles published in English were considered. Although significant contributions to climate change knowledge particularly from community organizations, governments, NGOs, and interdisciplinary academic efforts are often published in gray literature and review formats, these sources were excluded to maintain methodological consistency, ensure reproducibility, and focus on studies presenting original empirical data. Additionally, such documents are often less accessible, not consistently archived, and may be less likely to inform formal planning or decision-making processes, which tend to rely more heavily on peer-reviewed literature.

Papers were included if they (a) explored the effects of climate change in sub-Saharan Africa, (b) explored the impacts of climate change on health outcomes, (c) focused on relevant climate change impacts, including water quality, flooding, and drought and other health outcomes (d) were published between January 1, 2001, and August 1, 2024, and (d) were written in English. 2001 was selected as the starting point because it marks the beginning of the third assessment cycle of the Intergovernmental Panel on Climate Change (IPCC) [30], which significantly advanced the global discourse on climate change and its regional impacts. This period also aligns with a noticeable increase in peer-reviewed literature focused on climate-health linkages in SSA, making it a logical starting point for capturing the evolution of research in this field.

On the other hand, papers were excluded if they were not from the sub-Saharan Africa region or if they mentioned climate change impacts not relevant to the African context (e.g., changes in snowmelt, sea ice loss or glacial retreat). Of the 7851 identified studies, 921 were duplicates, and 6628 were excluded because they did not meet the inclusion criteria. A total of 302 studies were screened for eligibility. Of the 302 studies, 149 were excluded; 53 were reviews, 2 were animal health research, 44 were malaria research, 23 were opinion pieces and correspondences, and 27 studies had the wrong geographical settings, study design or outcomes. In total, 153 studies were included for review (Fig 1). Malaria-focused studies were excluded due to the large existing body of literature examining the relationship between climate change and malaria in SSA. Therefore, this effort was directed toward other health outcomes that have received comparatively less attention in the literature. To reduce selection bias, study screening was conducted independently by two reviewers.

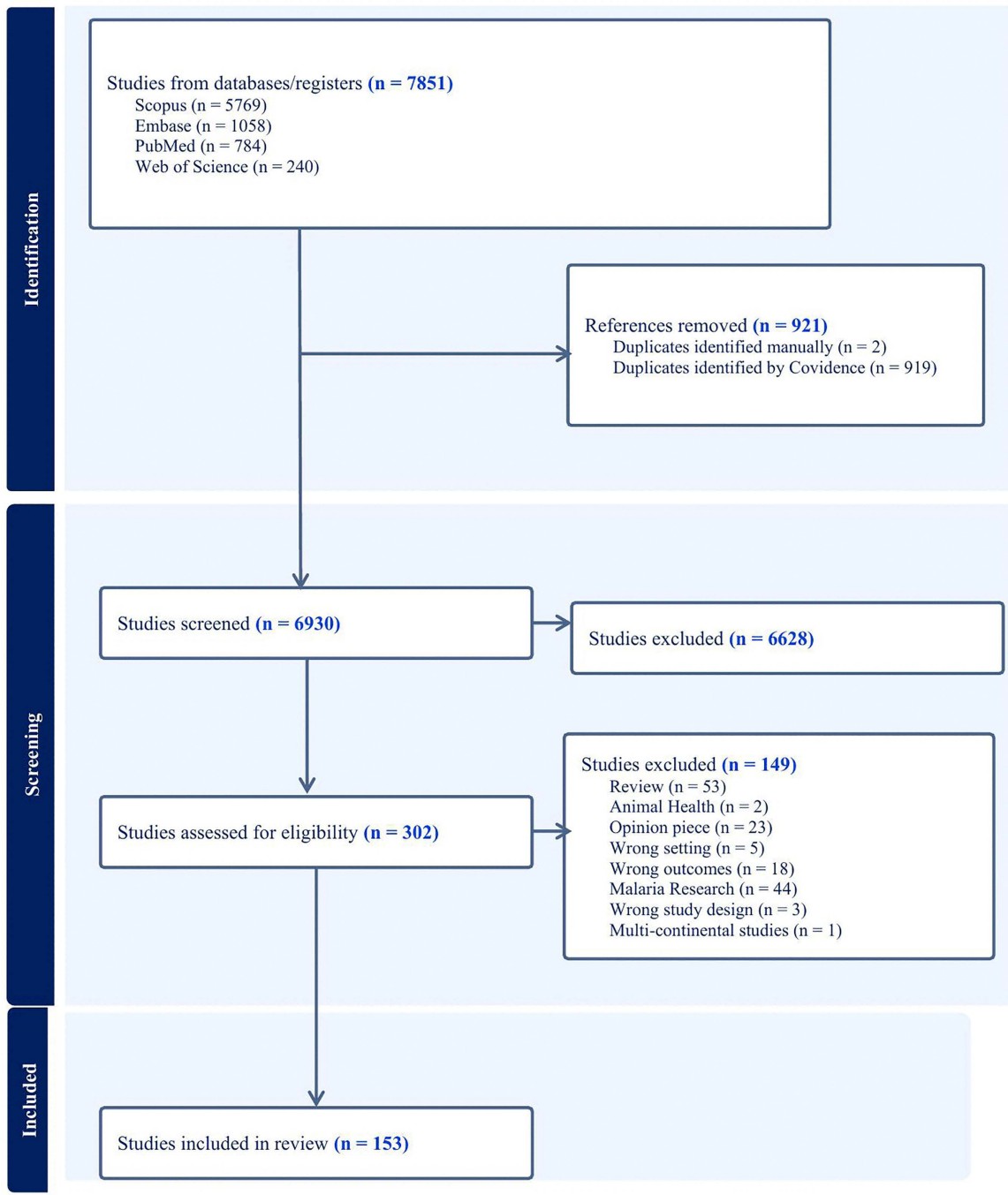

**Fig 1. PRISMA flowchart showing the number of articles identified in PubMed, screened and included in the scoping review.**

## Data extraction

Data extraction was conducted independently by two reviewers following the screening process. Each reviewer extracted key details from the included studies, including study characteristics (e.g., study design, population, country), climate change impacts discussed, health outcomes explored, and the types of interventions or solutions presented. Data were

extracted into a structured data charting form within the Covidence platform to ensure consistency and minimize errors. Following the extraction process, a verification step was conducted to ensure that all relevant data had been captured accurately and consistently. This step included comparing the data extractions of both reviewers for discrepancies. Any inconsistencies were discussed and resolved through consensus, with a third reviewer consulted if necessary.

### Data analysis

The extracted data were stored securely in Covidence and later exported into an spreadsheet for analysis. A hybrid thematic analysis was employed to synthesize findings from the included studies. An initial set of themes was developed a priori, informed by existing literature on climate change and health in sub-Saharan Africa, as well as the objectives of this scoping review. These preliminary themes included: highlighting research needs, informing science, providing solutions and knowledge to combat climate impacts, highlighting policy needs, and identifying resource needs. During the data charting process, these themes were iteratively refined through an inductive review of the literature to capture additional patterns and insights that emerged across the studies. This combination of deductive and inductive thematic analysis ensured a structured yet flexible synthesis of findings. To verify the robustness of the thematic analysis, two independent researchers cross-checked the categorization of studies into themes. Any discrepancies in theme assignment were discussed and resolved through consensus.

Additionally, the countries where previous climate change and health studies were conducted were visualized on a heatmap, providing insight into which countries have been studied extensively and where significant gaps in information exist. The thematic groupings enabled a comprehensive understanding of how climate change affects the region and its inhabitants. By categorizing the literature in this way, we could pinpoint specific gaps in research and policy that need to be addressed. The map further highlighted underresearched regions, emphasizing the need for more focused studies in those areas.

Finally, results were visualized in tables, charts, figures, and maps, enabling a clear presentation of findings. The analysis helped identify practical solutions and resources that can be mobilized to mitigate the adverse effects of climate change on health outcomes in SSA. This approach ensures that the findings contribute to academic knowledge and provide actionable insights for policymakers, researchers, and practitioners working to combat climate change and its impacts on public health.

## Results

The majority of the studies published on climate change and health in sub-Saharan Africa between 2001 and August 2024 focused on extreme heat (71 studies), extreme precipitation events (52 studies), drought (45 studies), and flooding (34 studies). Additionally, a moderate number of studies have focused on infectious diseases (24 studies) and microbial water quality (23 studies). For almost 24 years, only 10 studies have been published on climate change and air quality and 8 studies on chemical water quality and natural disasters (Table 1).

Of the 153 studies reviewed, 58 informed science by providing insight into climate trends and impacts; 43 studies discussed specific solutions and knowledge to help mitigate the impact of climate change on health and resource needs, 20 of the studies discussed and highlighted research needs, and 15 studies highlighted policy. Additionally, 11 studies explored multiple themes (Table 2).

Most studies included in this scoping review focused on water, sanitation and hygiene issues (n = 57), food security and malnutrition (n = 40), physical illness (n = 32) and health risks associated with pathogens (n = 26). The remaining 53 articles covered loss of livelihood due to natural disasters, climate induced displacement, mental health, gender-based violence, HIV and death (Table 3).

This scoping review also identified specific countries in sub-Saharan Africa where climate change and health research has been conducted, aiming to highlight countries with insufficient research. Result shows that most of the studies were

**Table 1. Climate Change Impacts in sub-Saharan Africa Identified by Past Studies.**

| Climate Impact | n | Studies |
|---|---|---|
| Extreme Heat | 71* | [4,10,11,14,18,19,31–95] |
| Extreme Precipitation | 52 | [4,14,18,19,32–36,38–41,46–50,52,54,57,64,68,69,77–79,81,83,86,90,91,93–112] |
| Drought | 45 | [2,14,34,36–49,97–103,113–132] |
| Flooding | 34 | [2,4,8,14,19,35,38,40–48,52,56,57,67,68,90,95,97,99–101,103,107,128,133–136] |
| Microbial Water Quality | 23 | [20,35,61,81,95,105,106,108,135,137–151] |
| Infectious Diseases | 23 | [1,11,14,47,60,61,91,102,108,138,143,151–161] |
| Air Quality | 10 | [31–35,96,113,162–164] |
| Chemical Water Quality | 8 | [35,137,139,141,142,145,146,165] |
| Natural Disasters | 8 | [8,16,41,42,47,67,94,100] |

\* Some studies covered multiple themes. Hence, N ≠ 153.

**Table 2. Research Categories and Number of Studies in Identified Categories from Previous Studies in Sub-Saharan Africa.**

| | n | References |
|---|---|---|
| Informs science – Provides insight for climate trends and impacts | 58 | [1,4,8,11,33,34,36,39,42,46,67,68,71,73,74,77–85,88–91,93,102,106–108,110–113,120,129,130,132,133,136,143,145–147,149–151,155,159–161,164,166,167] |
| Provides solution and knowledge to combat climate impacts | 43 | [16,35,43,49,50,53,54,56,66,69,76,86,94,99–101,105,115–117,119,125–127,138–142,144,148,152,154,157,162,168–170] |
| Highlight research needs | 20 | [19,44,45,55,57,59,60,63–65,72,75,97,104,109,122,124,127,128,163] |
| Highlights policy needs | 15 | [14,38,40,41,62,70,87,92,96,134,153,156,171] |
| Highlights resources needs | 6 | [48,52,95,114,123,172] |
| Multiple Categories in a single study | | |
| Highlight research needs; Highlights policy needs | 3 | [31,61,158] |
| Informs science – Provides insight for climate trends and impacts; Highlights resources needs | 2 | [98,103] |
| Informs science – Provides insight for climate trends and impacts; Highlights policy | 2 | [47,131] |
| Provides solution and knowledge to combat climate impacts; Highlights policy needs | 1 | [2] |
| Provides solution and knowledge to combat climate impacts; Highlights policy needs; Highlights resources needs | 1 | [137] |
| Provides solution and knowledge to combat climate impacts; Informs the state of climate science in Africa | 2 | [32,118] |
| Highlights policy needs: Highlights resources needs | 1 | [20] |

conducted in South and East Africa, with South Africa and Kenya having 24 and 22 published studies, respectively, followed by Ghana [14], Tanzania [10], and Burkina Faso [9]. Twenty-six studies included in this study were large scale studies conducted across multiple SSA countries (S1 Table). Only 24 out of 53 countries in sub-Saharan Africa have published studies on climate change and health over the past 24 years and 15 countries had less than 5 published studies on climate change and human health (Fig 2).

Research funding sources were also a focus of this study. Of the studies considered, 45% (n = 69) were funded by external grants, while 55% (n = 84) were self-funded by researchers, declared as "no external funding" in research publication (Fig 3). Of the 45% (n = 69) of the research funded by external grants, 88.4% (n = 61) were grants provided by international agencies across North America, Europe and Asia, including but not limited to German Academic Exchange Services

**Table 3. Climate Change and Health Issues in sub-Saharan Africa Identified by Past Studies.**

| Climate Change Impact | N* | Studies |
|---|---|---|
| WASH issues | 57 | [1,2,11,14,16,18–20,33,34,36,38,43,44,52,56,66,81,86,91,93,94,101, 104–109,113,115,122,135,137,139–144,146,148,149,151,158,167,170] |
| Food security/Malnutrition | 40 | [4,11,14,31,35,38–40,43,44,46,48,52,60,75,98,99,101,102,111,113,114,116–121,123,124, 126–128,132,139,142,147,166,168,172] |
| Loss of livelihood | 8 | [45,49,52,97,117,125,133,166] |
| Death | 15 | [10,50,57,58,60,62,63,71,72,80,82,84,88,92,163] |
| Other pathogens | 26 | [14,31,33,38,44,52,54,56,60,61,78,79,83,90,101,105,107,110,124,142–144,150,153,161] |
| Physical illnesses | 32 | [1,11,34,37,49,53–56,58,68,70,72–74,76,77,85,92,94,107,113,115,134,139,142,153,160,164,166,173] |
| Climate-induced displacement | 6 | [40,41,47,100,123,124] |
| Mental health (not limited to anxiety and depression) | 13 | [11,14,31,42,56,67,97,114,123,134,136,139,166] |
| Social determinant of health (including but not limited to social identity, attachment to place, safety, well-being) | 7 | [8,52,87,89,97,133,139] |
| Gender-based violence | 2 | [8,40] |
| HIV/AIDS | 2 | [68,131] |

\* Some studies covered multiple themes. Hence, N ≠ 153.

(DAAD), the Wellcome Trust, the Taiwan Ministry of Science and Technology Grant, the World Health Organization, the United States National Science Foundation, the Public Health Agency of Canada, the National Institutes of Health, the Rockefeller Foundation, the International Climate Change Information and Research Programme and Commonwealth. Funding was also provided by universities in the Global North, such as the University of Guelph. Only seven studies (11.6%) were funded through local funding provided by government agencies of sub-Saharan countries; four of those were provided by the South African Medical Research Council, one was a grant from Fundo Nacional de Investigação (FNI) in Mozambique, one was a grant from the Ethiopian Institute of Water Resources and one from the African Climate Change Fellowship Program (ACCFP).

Additionally, this study explored the impact of climate change on vulnerability and the health of vulnerable groups (S1 Fig). Vulnerable groups, such as children, the elderly, pregnant women, low-income populations, and those with pre-existing health conditions, are disproportionately affected by climate change due to factors like limited healthcare access, poor infrastructure, and heightened exposure to climate risks. Among the studies reviewed, 59 did not consider vulnerability in their research. Of those who did, 40 focused exclusively on children, while 23 addressed children along with other vulnerable groups, such as elderly individuals, pregnant women, disabled individuals, low-income groups, and immunocompromised individuals. Thirteen studies specifically examined gendered vulnerabilities, with 12 studies focused on women and 1 on men. Five studies exclusively focused on immunocompromised groups, 3 studies investigated the impacts of climate change on the elderly population, and an additional 2 investigated the effects on impoverished populations. Three studies focused on pregnant women, 1 study on farm workers, and 2 studies focused on mining workers. Finally, one study examined the impacts of climate change on the health of internally displaced populations and on local and Indigenous communities.

This research also explored the various adaptation strategies proposed to mitigate climate change impacts on SSA, as suggested by the authors of the included studies (S2 Fig). Thirty-seven studies (24%) had no proposed adaptation strategies. Adaptation strategies were grouped into themes including technology gaps, policy modifications, community-focused adaptation plans, climate preparedness programs and the need for funding. The majority of studies proposed a community-focused adaptation plan and the inclusion of local context in climate intervention (n = 37, 27%) as the most

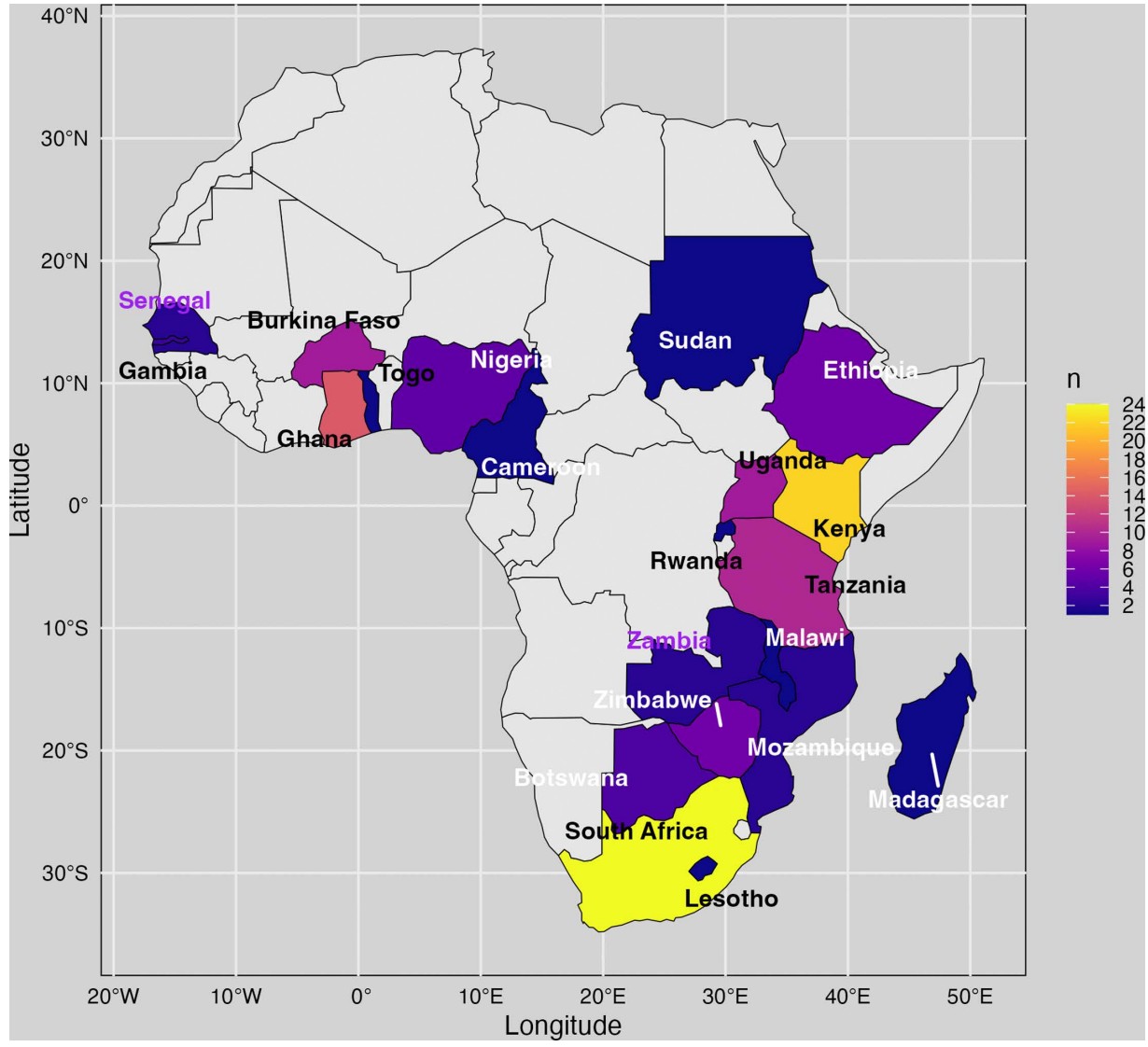

**Fig 2. Map showing the number of climate change and health studies in sub-Saharan Africa, color-coded by research volume.** Purple to yellow shades indicate more studies (2 to 24), while gray countries had no studies meeting the inclusion criteria (2001–2024). Only English-language publications were included.

effective adaptation strategy to help residents of SSA minimize the impacts of climate change. Improved technology was the second highest proposed adaptation strategy suggested by the studies reviewed (n = 32, 23%), and the authors proposed technological improvements such as early warning systems, remote sensing, robust meteorological technologies, water treatment systems, vaccination, disease surveillance systems, energy-saving air conditioning, and climate-smart agricultural technologies. A total of 18% (n = 25) of the studies discussed inadequate local funding as a significant barrier to climate change adaptation and proposed an increased budget for climate-related concerns. Similarly, an additional 18% (n = 25) of the studies highlighted the need for climate preparedness programs among citizens, government officials and health workers. Finally, 14% (n = 20) of the studies further describe policy modification, specifically suggesting holistic and concerted efforts by individuals, multiple government agencies working together and donor agencies in the fight against climate change and resulting health impacts.

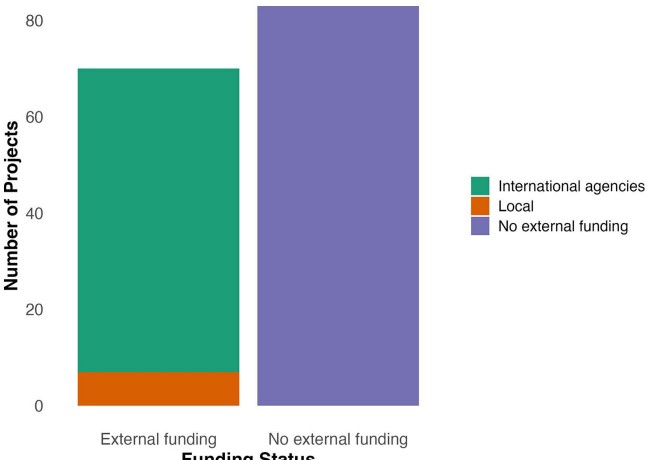

**Fig 3. How are climate studies funded in sub-Saharan Africa.**

## Discussion

This study aimed to understand the relationship between climate change and health in SSA. Many of the events documented in the literature; such as seasonal droughts, floods, and heat episodes are part of long-standing and established weather patterns in the region, rather than direct consequences of anthropogenic climate change. However, these weather-health relationships are critically important for informing climate change adaptation strategies. As climate change is projected to increase the frequency, intensity, and unpredictability of extreme weather events, current health burdens linked to seasonal weather patterns can serve as proxies for anticipating future risks and vulnerabilities. The impact of climate change is already apparent in SSA, as the research reviewed in this study indicates that regions is already being affected by drought [60,113,118,125], extreme heat [9,53,57,59,63,166] and microbial and chemical water quality issues, leading to infectious diseases such as cholera and diarrhea outbreaks as a result of compounding climate extremes on water access and quality [9,16,56,135,140,146,174]. Additionally, there have been reports of natural disasters, such as floods [8,44,134,146] and cyclones [16], all contributing to negative health outcomes in addition to economic costs.

Research on climate change and health is critically important in SSA because the region faces heightened vulnerability due to a combination of environmental, social, and economic factors. Many countries in SSA are already experiencing the effects of climate variability that directly and indirectly impact human health [175]. At the same time, limited infrastructure, under-resourced health systems, high disease burdens, and widespread poverty constrain the capacity to respond to these challenges [176]. Understanding the intersection of climate and health in this context is essential for identifying risk pathways, informing adaptation strategies, and strengthening resilience.

The intensified climate events on the continent are leading to several physical, mental and emotional health impacts. Most of the studies in this review focused on WASH issues, food security, physical illnesses such as heat exhaustion, and pathogen infections such as cholera and malnutrition. However, the social, emotional and mental aspects of climate change have been relatively understudied. Loss of livelihood due to the impact of floods, extreme temperature and drought on agricultural, resource-based occupation and outdoor informal economies has been reported in a few studies in the sub-Saharan African context [52,133]. Loss of livelihood is often intertwined with disrupted community ties, loss of identity, climate-induced displacement, climate anxiety, depression and gender-based violence [40,41,52,123,166]. Therefore, climate change is leading to cascading outcomes affecting all aspects of health and wellness in the region.

There is an urgent need for accelerated climate change research in Africa, home to more than 1 billion people and experiencing rapid population growth. Despite the critical need for solutions, only 43 of the studies reviewed in this study

discussed specific strategies and knowledge to mitigate the impact of climate change on health. These studies included contributions to new research, evaluations of existing mitigation efforts, and insights into issues unique to the continent, such as lack of access to basic services and nutrition. Several studies have provided specific mitigations, considerations, and recommendations. For example, a study in Ethiopia examined childhood diarrhea, finding a high incidence at the beginning of dry seasons [36]. Other studies have addressed climate-specific issues such as extreme temperatures, precipitation, and mortality and proposed mitigation steps to reduce hospital admissions, morbidity, and mortality [50,53,54], including within-hospital settings. The challenges related to WASH were also highlighted, with some studies identifying opportunities for integrating climate adaptation into WASH development planning [2]. The findings and recommendations from these studies could be implemented in other regions facing similar challenges. Research on water burdens and gender inequalities has emphasized specific strategies for reducing these disparities [115]. Overall, these studies offer valuable insights and recommendations that are specific to the region, that can be applied to mitigate the impact of climate change on health across SSA.

This review revealed key gaps and opportunities in climate change and health research in SSA. First, there are significant geographic disparities, with some countries being more frequently studied than others. This uneven distribution of research limits the ability to develop comprehensive, country-specific interventions and may lead to an incomplete understanding of regional needs. Additionally, while some studies adopt a multi-country or regional approach, aggregated data can obscure critical local disparities [177]. SSA is not a monolith; differences in climate zones, infrastructure, governance, and health system capacity create diverse risk profiles both across and within countries. Tailored, context-specific research is essential for designing effective adaptation strategies that reflect the unique vulnerabilities and strengths of individual communities [178]. Recognizing and addressing these intra-regional differences is key to promoting equitable and resilient responses to climate change. Furthermore, even in countries where research has been conducted, several authors emphasize the need for more primary climate and health data to support evidence-based decision-making [14,128,171]. Expanding research efforts to include understudied countries can provide a more comprehensive understanding of climate change and health impact across SSA, ensuring that attention is given to those most impacted by climate change and the opportunity to learn from each other.

There are key considerations that need to be taken into account when discussing climate change solutions for SSA. For instance, we need to integrate and discuss social determinants of health as we think of climate change and health. Only five studies highlighted or discussed the associations between health, social determinants of health, and climate change [8,52,97,133,139]. However, it is well understood that the work a person does, where people live, their educational level, and income level all play a role in determining health [179]. More research is needed to account for climate change as a determinant of health, especially since low-income countries will be impacted the most. Additionally, there are only a few studies on how climate change leads to gender-based violence. However, relationships between climate change, loss of livelihood, feelings of emasculation and gender-based violence have been reported in sub-Saharan Africa [8,40].

Many studies have not adequately considered the vulnerability of different population groups through an intersectional lens. While several papers acknowledged groups such as children, pregnant women, and the elderly as vulnerable, these mentions were often superficial, and 38 studies did not consider vulnerability at all. This gap highlights a pressing need for more comprehensive research that accounts for the diverse experiences and needs within communities. Climate change does not impact all people equally and its effects are shaped by existing social, economic, and environmental inequalities [180]. Understanding community heterogeneity and the overlapping vulnerabilities faced by marginalized groups is essential for developing effective adaptation and mitigation strategies. Policies must be revised to move beyond a one-size-fits-all approach and instead prioritize equity by identifying and addressing the specific needs of those most at risk. This includes integrating data on gender, age, income level, disability, and social exclusion into climate-health planning [180]. Incorporating vulnerability ensures that resources are allocated more fairly, that interventions are more effective, and that the resilience of entire communities is strengthened and not just the most visible or privileged segments. Ultimately,

placing vulnerability at the center of climate-health policy is not only a matter of justice but also a practical necessity for sustainable and inclusive development in SSA.

Further, while some climate themes have been extensively studied, others remain underexplored and require greater attention. There is a notable lack of research in areas such as extreme heat, natural disasters, and air pollution. For instance, only 10 studies highlighted research related to air pollution [31–33,96,113,162,163], and just two specifically addressed the intersection of climate change and cardiovascular diseases [96,115]. This is a significant gap, especially considering that the Lancet Global Burden of Disease (2016) reported that 33% of the global burden of stroke is attributed to environmental factors such as air pollution and lead exposure [181]. Air pollution contributes to climate change through the release of greenhouse gases and short-lived climate pollutants, but it can also be worsened by climate-related phenomena such as wildfires or stagnant air masses during heatwaves. This study also revealed that air pollution has emerged as a significant contributor to the global burden of stroke in low- and middle-income countries (LMIC) [181]. Climate change is expected to increase drought and air pollution in some regions of the world, including SSA [182]. Despite this significance, research in this area is almost nonexistent and requires urgent attention.

A few other studies offer concrete pathways and recommendations on what countries need to focus on and prioritize to reduce the impact of climate change. Technological advancement was one of the most common emerging themes from the included studies, although implementing the suggested technologies might be far-fetched in SSA landscapes. For example, in areas where infectious diseases are a concern, authors recommend focusing on early warning signs, disease forecasting, and climate disaster mitigation [14,128] to strengthen public health systems to cope with and reduce climate change impacts. However, it is difficult in low-income countries to focus on early warning systems or disease forecasting when basic solutions for preventing diseases, such as access to safe and clean water or sanitation, are still lacking.

For instance, without the additional burden of intensified climate impacts, current water and sanitation infrastructure or healthcare infrastructure is inadequate to protect public health [2,17,19,104,140]. Some countries are already experiencing preventable waterborne diseases that could be addressed by having adequate water and sanitation infrastructure. Some examples include the cholera outbreak in Tanzania in 2023, ongoing in 2024, and another in June 2024 in Nigeria [16,18,178–180], both attributed to climate change. In addition, lack of technologies and inadequate electricity coverage create barriers to implementing other simple solutions to combat extreme heat conditions [166] and a lack of improved agricultural technologies to combat food insecurity [4]. For example, irrigation systems are creating vulnerabilities further intensified by climate change. The continent must catch up to address current poor living conditions and, at the same time, accelerate the research needed to move to the 21st century to address climate change issues.

In tandem with inadequate climate policies, a lack of funding was also identified as limiting climate change research progress in SSA. In the present study, more than half of the studies reviewed were self-funded by researchers. The large amount of self-funded research highlights the financial constraints researchers face, which can limit the scope, scale, and depth of their studies, ultimately hindering the development of comprehensive climate change mitigation and adaptation strategies. Additionally, only seven of the 153 selected studies were funded by local governmental agencies, revealing a much deeper systemic issue about the perceived relevance of climate change impacts in this region. Therefore, there is an urgent need for increased investments in climate research [2,19,20,52,101,114,127]. Funding climate research will help inform climate decision-making and policies [128]. There is also a lack of climate consideration in government policies [2]. To ensure that climate investment and increased funding reap the maximum benefits, there is a need for multiagency collaboration between government agencies to inform policies that achieve economic, environmental, social and health justice [97,173]. Local residents and external donor agencies should also be included in climate strategizing to ensure maximum benefits [68]. Addressing these limitations through enhanced funding, comprehensive policies, and inclusive collaboration will be pivotal for effectively mitigating the impacts of climate change in SSA.

Even with limited resources, countries can strategize and prioritize actions towards climate change. One thing is evident: the issue is not a lack of understanding of the urgency to address climate change. Studies evaluating climate

change perception in various African countries have shown that most of the population is familiar with climate change and have experienced it to some extent [14,41,100,125,127,183]. However, there is insufficient data due to technological and resource limitations to provide tailored interventions at the community and national, and lack of national policies. For instance, in Zimbabwe, following the devastation caused by Cyclone Ida, natural disasters exposed another problem—food insecurity. A study of 19 health facilities conducted after Cyclone Ida revealed that 94% of the facilities were not equipped to address malnutrition, either due to lack of proper training, inadequate staffing, or failure to follow proper protocols [126]. With climate change, areas with limited resources will be most affected, and there is a need for adequate systems to address climate change and health issues. Therefore, community-focused adaptation plans are necessary, taking into account the strengths and vulnerabilities of individual communities and countries. These details can only be revealed through community-level or national research, which is often obscured by large-resolution, aggregated datasets produced by major development agencies.

There is also a dire need to build capacity for health sector personnel through climate preparedness programs that integrate climate resilience into public health systems. Educational and research institutions can offer expertise in addressing regional or national climate change solutions. These programs should focus on enhancing the ability of healthcare infrastructure to withstand climate-related events and training healthcare professionals to respond effectively to climate-induced health issues [6,38,100,107,147,182,184]. Additionally, community-based interventions are essential for educating and empowering local populations on climate-related health risks and adaptive practices. Incorporating climate education in formal and informal places of learning in communities, has shown to increase ability to identify vulnerability and build adaptive capacity [38,42]. These climate education programs may include utilizing traditional knowledge to inform strategies to reduce individual- and community-level vulnerabilities, improve infrastructure, create multiple climate-resilient income streams, and address the intersectional vulnerabilities that climate change exposes [42]. Due to the large population of young people in Africa, strategies targeting young people are paramount [41]. Educational programs may also target improved monitoring and surveillance systems [14,171] to fill the data gap and improve data-driven decision-making. These efforts will ensure that public health systems are resilient, adaptive, and equipped to protect the well-being of all individuals, especially the most vulnerable populations.

Although this paper has several merits, including the identification of key interconnections between climate and health in sub-Saharan Africa, it is not without limitations. One major limitation is the inclusion criterion of English-language publications only. This introduces a language bias that may underrepresent research conducted in Francophone, Lusophone (Portuguese-speaking), and Arabic-speaking countries. As a result, these regions may be incorrectly characterized as lacking in climate and health research, when in fact relevant studies may exist but were excluded due to language barriers. Additionally, this review excluded studies focused solely on malaria due to the extensive and well-established body of literature on the subject. While this decision allowed for the exploration of underexamined health outcomes, it may also omit key climate-sensitive disease trends that remain highly relevant to public health in the region.

## Conclusion and recommendations

This scoping review reveals a substantial and growing body of literature addressing the intersection of climate change and health in SSA, but it also highlights significant gaps in research, policy implementation, and resource allocation. While there has been progress in understanding climate-related health outcomes—particularly regarding WASH, food security, and infectious diseases—other critical issues, such as mental health, gender-based violence, and the health impacts of air pollution and extreme heat, remain severely underexplored. The uneven geographic distribution of studies, predominance of self-funded research, lack of localized data, and limited attention to vulnerable populations reflect systemic constraints that impede effective climate adaptation and public health response in SSA. Despite widespread awareness of climate change impacts among local populations, there remains a critical need for actionable data, tailored adaptation strategies, and sustainable institutional support to protect public health in the face of a changing climate

To address these challenges, we recommend the following: 1) Expand geographic and context-specific research by prioritizing funding in understudied countries and tailor research to local vulnerabilities; 2) Invest in capacity building by integrating climate training into public health and medical education and establish preparedness programs for health professionals; 3) Increase local and national funding by encouraging national governments to fund climate-health research and reduce reliance on external donors; 4) Mainstream vulnerability and equity by adopting intersectional approaches in policy and research, to address the needs of marginalized groups; 5) Strengthen data and surveillance systems to enhance real-time data collection and support evidence-based policy-making tailored to local contexts; 6) Integrate climate considerations into development planning by embedding them into national health, agriculture, and disaster response agendas; and 7) Support community-led adaptation by leveraging traditional knowledge locally driven sustainable strategies.

Implementing these recommendations will support SSA in developing inclusive, equitable, and resilient climate-health strategies essential for long-term public health protection.

## Supporting information

**S1 Fig. Vulnerable Groups Identified in Past Studies and the Number of times the group was mentioned.** (TIF)

**S2 Fig. Adaptation Strategies Suggested by Authors in Current Study to Address Climate Change and Health Issues in Sub-Saharan Africa.** (TIF)

**S1 Table. Number of Studies on Climate Change and Health by Country in Sub-Saharan Africa.** (DOCX)

**S1 Checklist. PRISMA 2020 checklist October 2025.** (DOCX)

## Author contributions

**Conceptualization:** Aminata Kilungo, God'sgift Chukwuonye, Victor Okpanachi.

**Formal analysis:** Aminata Kilungo, God'sgift Chukwuonye, Victor Okpanachi.

**Methodology:** Aminata Kilungo, God'sgift Chukwuonye, Victor Okpanachi.

**Supervision:** Aminata Kilungo.

**Visualization:** Aminata Kilungo, God'sgift Chukwuonye.

**Writing – original draft:** Aminata Kilungo.

**Writing – review & editing:** Aminata Kilungo, God'sgift Chukwuonye, Victor Okpanachi, Hussein Mohamed.

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
