## [Decision Letter · Decision Letter 0]

13 Feb 2025

Assessing Sub-Saharan Africa's Readiness to Address the Impact of Climate Change and Health: A Scoping Review

PLOS ONE

Dear Dr. Kilungo,

Thank you for submitting your manuscript to PLOS ONE. After careful consideration, we feel that it has merit but does not fully meet PLOS ONE’s publication criteria as it currently stands. Therefore, we invite you to submit a revised version of the manuscript that addresses the points raised during the review process.

https://journals.plos.org/plosone/s/submission-guidelines#loc-laboratory-protocols . Additionally, PLOS ONE offers an option for publishing peer-reviewed Lab Protocol articles, which describe protocols hosted on protocols.io. Read more information on sharing protocols at https://plos.org/protocols?utm_medium=editorial-email&utm_source=authorletters&utm_campaign=protocols .

We look forward to receiving your revised manuscript.

Kind regards,

James Colborn

Academic Editor

PLOS ONE

4. We note that Figure 2 in your submission contain [map/satellite] images which may be copyrighted. All PLOS content is published under the Creative Commons Attribution License (CC BY 4.0), which means that the manuscript, images, and Supporting Information files will be freely available online, and any third party is permitted to access, download, copy, distribute, and use these materials in any way, even commercially, with proper attribution. For these reasons, we cannot publish previously copyrighted maps or satellite images created using proprietary data, such as Google software (Google Maps, Street View, and Earth). For more information, see our copyright guidelines: http://journals.plos.org/plosone/s/licenses-and-copyright.

5. We notice that your supplementary figure and table are included in the manuscript file. Please remove them and upload them with the file type 'Supporting Information'. Please ensure that each Supporting Information file has a legend listed in the manuscript after the references list.

6. Please remove your figures from within your manuscript file, leaving only the individual TIFF/EPS image files, uploaded separately. These will be automatically included in the reviewers’ PDF.

Reviewers' comments:

Reviewer's Responses to Questions

**Comments to the Author**

1. Is the manuscript technically sound, and do the data support the conclusions?

Reviewer #1: Partly

Reviewer #2: Yes

2. Has the statistical analysis been performed appropriately and rigorously?

Reviewer #1: N/A

Reviewer #2: Yes

3. Have the authors made all data underlying the findings in their manuscript fully available?

Reviewer #1: Yes

Reviewer #2: Yes

4. Is the manuscript presented in an intelligible fashion and written in standard English?

Reviewer #1: Yes

Reviewer #2: Yes

Reviewer #1: Thank you for the opportunity to review this manuscript. The authors conduct a timely scoping review of climate change and health research in Sub-Saharan Africa and present the gaps and recommendations for researchers and policy makers to address these.

Abstract

- Add the word “studies” to the 8 behind natural disasters.

Introduction

- Curious about the use of the terms “developing countries”. There has been a lot of discourse around this language. Most articles now use low-income countries or regions.

- Remove this sentence – it repeats the same information above “Some countries are already experiencing higher rates than normal from malnutrition, diarrhea cholera, and vector-borne diseases such as dengue (21).”

- In the Research Questions why in #3 are you restricting strategies to air and water quality, extreme heat, enhancing resilience and mitigating health impacts? I would suggest instead “What practical solutions and community-based adaptation strategies can be developed and implemented to enhance resilience and mitigating health impacts to climate change ?”

Methods

- I would remove this: Climate change impacts relevant to sub-Saharan Africa include extreme heat, food security, microbial and chemical water quality, flooding, and drought. There are more impacts than this and the language is not consistent.

- You should include reasoning for this “No white papers, gray literature, review papers, or other sources were included.”, and only those written in English. A lot of community, government, NGO and academic research on climate change is published as these types of documents. You can make the argument though that these types of documents are less accessible and may be less likely to be used in planning as result.

- Why was 2001 considered your starting point?

- I would rephrase this “(a) addressed the conceptualization of climate change in sub-Saharan Africa” to “Explores the effects of climate change in sub-Saharan Africa”

- Why would wildfires not be relevant to the African context? This is not accurate: Africa’s lush tropical forests face a surprising threat: fire,

- Again zoonotic impacts of climate change are of large concern, better rationale for the exclusion of the malaria research is required.

- The description of thematic analysis needs to be expanded, it is unclear if the themes were selected apriori based on the literature OR if they were developed based on the research included in the literature review. IF it was the later then the “themes” would be results and should not be presented in the methods. If they were selected PRIOR to analysis/extraction of the included articles then rationale for how these were selected is required.

- No extraction of data from the articles is described only the screening. Was the extraction also conducted by 2 people, was it done simultaneously with the screening? Please add more details.

- How was data stored/analysed/verified?

Results

- These should be re-ordered in terms of magnitude: The majority of the studies published on climate change and health in sub-Saharan Africa between 2001 and August 2024 focused on extreme heat (71 studies), drought (45 studies), extreme precipitation events (52 studies) and flooding (34 studies).

- You use studies and publications interchangeably, please choose one.

- I find lines 244-250 confusing. The way it is phrased sounds like a comparison. I would rephrase as: Most studies included in this scoping review focused on water, sanitation and hygiene issues (n=57), food security and malnutrition (n=40), physical illness (n=32) and health risks associated with pathogens (n=26). The remaining 53 articles covered loss of livelihood due to natural disasters, climate induced displacement, mental health, gender-based violence, HIV and death (Table 3).

- “This scoping review also examined specific countries in sub-Saharan Africa where climate change and health research originated.” Did you mean originated or took place? Originated implies it was conceived and conducted locally which is a great aspect to asses but it does not appear you did this.

- I would move this to the figure description: “The findings were visualized on a map of Africa, with color coding from 4 to 24 studies, where a dark hue represents countries with more research and a lighter hue indicates countries with fewer studies. Countries shown in gray represent those with no published studies on climate change and health between 2001 and August 2024 that fit our criteria.”

- I think there is a missed opportunity in the section on vulnerability on what makes these groups vulnerable and if there are any commonalities between these groups, particularly how policies should be leveraged for these groups.

- Capitalize Indigenous.

- Pie charts don’t provide much value, I would consider another format or remove the figure all together.

- Add the reference in Table 1 in the row “study across multiple countries in SSA” to their respective countries and add a * to indicate it covers additional countries.

- There are a number of issues with the tables and figures please verify the numbers:

o The number of entries in the supplementary figure 1 don’t add up to 153.

o There are more entries than studies in Table 1 “Table 1. Number of Studies on Climate Change and Health by Country in Sub-Saharan Africa”

Discussion

- Throughout the manuscript extreme weather is associated with climate change, however those two are not equivalent. For example, much of SSA experiences seasonal drought/dry seasons and rainy seasons. But, studies on these relationships between weather and health can be used to inform projections and impacts of climate change. More nuance is needed in the discussion.

- The argument for regional data is poor. It is important but the discussion presented here doesn’t illustrate why.

- You continue presenting results in the discussion section. Please review and then build off the results and link to the broader literature.

- Unclear why the malaria papers are brought up in the discussion when they were excluded from the review.

- I find the authors are confusing environment with being synonymous e.g. this is a significant finding, mainly since the Lancet Global Burden of Disease (2016) reported that 33% of the global burden of stroke is attributed to environmental factors such as air pollution and lead exposure (177). <<- does climate change cause air pollution OR is climate change caused by air pollution. I think some of these directionality questions require some nuance. Lead exposure can increase with climate change but is more indirect as a result of historical leaded gasolines and poor safety regulations on lead use in production.

- It is unclear how the recommendations were developed from the themes/results of this scoping review.

Conclusions and recommendations

- Your scoping review is on SSA – why are you referring to research relevant to the continent?

Reviewer #2: The authors have articulated with scientific rigour gaps in research on climate and health and provided for aligned recommendations as per the identified gaps in research and implementation. Minor revisions are however required under the following sections outlined below;

Abstract and introduction: Authors have contextualized the study well in the abstract and introduction however, the abstract does not mention the approach used in inclusion of articles “PRISMA”

Methods:

Rational for classifying articles as focused on infectious diseases and the exclusion of 44 other research as Malaria research; Consider adding supplemental file on the infectious diseases covered and if there is any recommendation that stands out to mitigate the effects of climate change on the disease. additionally, Rationale for this exclusion to be expanded or further clarified; Malaria transmission is a potential health outcome that is affected by climate change/resilience.

Figures & Tables

Figure 2 Title is not explicit mentioning the main bias of including only English written publication thus areas of Francophone, Portuguese and/or Arabic may be incorrectly classified here as lacking in climate and heath research.

Review of figure 3 to align with findings provided: No external funding n=83 as per line 275: Additionally, graph does not highlight all findings presented in discussion argument from line 283-287: Self-funded research and locally/domestically funded research has been highlighted as main finding

Final recommendation on line 495 to show limitations of the study and this paucity/gaps in publications is in for those done in English across the continent

**Do you want your identity to be public for this peer review?** For information about this choice, including consent withdrawal, please see our Privacy Policy

Reviewer #1: No

Reviewer #2: **Yes: ** Sadiq Kuto Wanjala

---

## [Author Response · Author response to Decision Letter 1]

18 Aug 2025

We have attached the reviewers comments. Please see cover letter with this submission.

---

## [Decision Letter · Decision Letter 1]

16 Oct 2025

Assessing Sub-Saharan Africa's Readiness to Address the Impact of Climate Change and Health: A Scoping Review

PONE-D-24-45324R1

Dear Dr. Kilungo,

We’re pleased to inform you that your manuscript has been judged scientifically suitable for publication and will be formally accepted for publication once it meets all outstanding technical requirements.

Kind regards,

James Colborn

Academic Editor

PLOS ONE

Additional Editor Comments (optional):

Reviewers' comments:

Reviewer's Responses to Questions

**Comments to the Author**

Reviewer #1: All comments have been addressed

2. Is the manuscript technically sound, and do the data support the conclusions?

Reviewer #1: Yes

3. Has the statistical analysis been performed appropriately and rigorously?

Reviewer #1: Yes

4. Have the authors made all data underlying the findings in their manuscript fully available?

Reviewer #1: Yes

5. Is the manuscript presented in an intelligible fashion and written in standard English?

Reviewer #1: Yes

Reviewer #1: (No Response)

**Do you want your identity to be public for this peer review?** For information about this choice, including consent withdrawal, please see our Privacy Policy

Reviewer #1: No

---

## [Editor Report · Acceptance letter]

PONE-D-24-45324R1

PLOS ONE

Dear Dr. Kilungo,

I'm pleased to inform you that your manuscript has been deemed suitable for publication in PLOS ONE. Congratulations! Your manuscript is now being handed over to our production team.

Kind regards,

on behalf of

Dr. James Colborn

Academic Editor

PLOS ONE